# Fair GANs through model rebalancing for extremely imbalanced class distributions

## Abstract

Deep generative models require large amounts of training data. This often poses a problem as the collection of datasets can be expensive and difficult, in particular datasets that are representative of the appropriate underlying distribution (e.g. demographic). This introduces biases in datasets which are further propagated in the models. We present an approach to construct an unbiased generative adversarial network (GAN) from an existing biased GAN by rebalancing the model distribution. We do so by generating balanced data from an existing imbalanced deep generative model using an evolutionary algorithm and then using this data to train a balanced generative model. Additionally, we propose a bias mitigation loss function that minimizes the deviation of the learned class distribution from being equiprobable. We show results for the StyleGAN2 models while training on the Flickr Faces High Quality (FFHQ) dataset for racial fairness and see that the proposed approach improves on the fairness metric by almost 5 times, whilst maintaining image quality. We further validate our approach by applying it to an imbalanced CIFAR10 dataset where we show that we can obtain comparable fairness and image quality as when training on a balanced CIFAR10 dataset which is also twice as large. Lastly, we argue that the traditionally used image quality metrics such as Frechet inception distance (FID) are unsuitable for scenarios where the class distributions are imbalanced and a balanced reference set is not available.

## 1 Introduction

Recent advances in generative models such as GANs and diffusion models have allowed them to generate highly realistic images. However, popular publicly available generative models are unfair (Maluleke et al., 2022; Jain et al., 2023). Fairness in generative models is defined as equal representation with respect to one or more attributes (e.g. ethnicity or gender) (Hutchinson and Mitchell, 2019). This implies that a generative model should be equally likely to generate images belonging to any of the classes in the training distribution. For example, a fair model with regard to ethnicity should generate images of a black and white individual with equal likelihood. The biased nature of SOTA generative models can be seen when randomly sampling 10,000 samples from the StyleGAN model. You get over 5,000 samples belonging to the white population with less than 500 samples for blacks and Indians combined. Similarly, the super-resolution model PULSE (Menon et al., 2020) generates only faces with lighter skin tones regardless of the demographic group of the input.

Recently, researchers have shown interest in using synthetic data generated by machine learning models either as augmentation or on its own for training various downstream classification tasks (Grosz and Jain, 2022; Jain et al., 2022; Jahanian et al., 2021; Jaipuria et al., 2020). The usage of synthetic data has shown various advantages, the most important of which is its ability to eliminate the need for privacy-sensitive datasets (e.g. medical imagery, biometrics). In fact, various countries have banned or regulated the use of biometric datasets including facial image datasets (Voigt and Von dem Bussche, 2017; McCarthy, de la Torre, 2018). Various popular biometric datasets were also withdrawn due to such issues (Harvey, 2021). However, models trained on synthetic data will be biased if the models generating the data are themselves biased. This can have major societal implications and can reinforce social biases such as in the case of face recognition systems that are widely used by governments and border patrols for security.

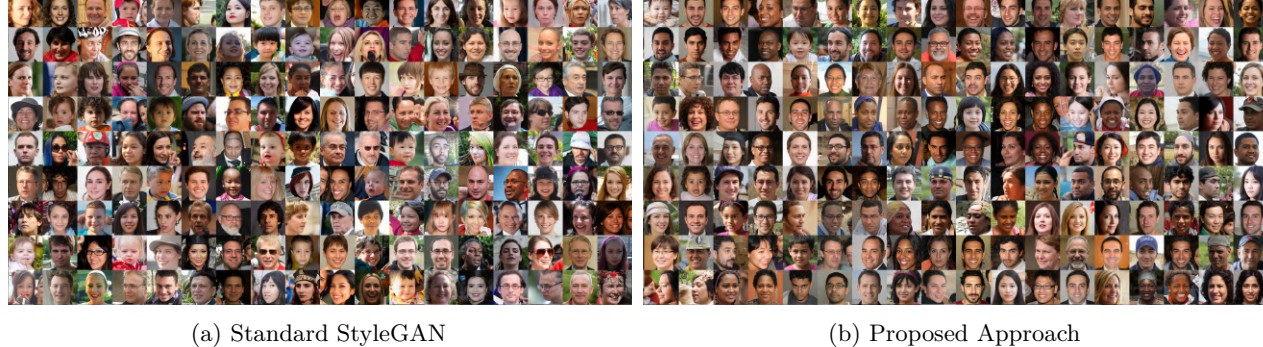

(a) Standard StyleGAN                    (b) Proposed Approach

Figure 1: Examples of images generated randomly using the standard StyleGAN2 and the proposed approach.

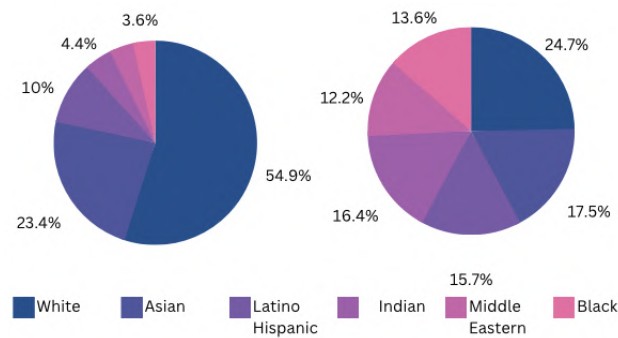

Figure 2: Pie charts showing the percentage of images belonging to different demographic groups when randomly generating 100,000 samples from the StyleGAN model trained on the FFHQ dataset (left) and our proposed approach (right).

Previously, researchers who studied de-biasing GANs assumed access to either a smaller balanced dataset (Teo et al., 2023) or a reference dataset from the same training data distribution (Choi et al., 2020; Grover et al., 2019) or a labeled dataset (Xu et al., 2018; Sattigeri et al., 2018). With generative models being trained in a self-supervised manner on datasets on the scale of billions of samples, these are impractical assumptions. There are various applications such as medical imagery and biometrics where balanced/ labeled datasets are difficult to collect and do not exist for most cases, let alone one with the same data distribution.

In this work, we do not assume access to either a labeled or a reference dataset. In fact, we make an even stronger assumption that we do not have access to the original training dataset altogether. Access to the original training dataset is often problematic as it relies on access to a privacy-sensitive dataset. There are also various applications in which the original data used for training a GAN model is either no longer available or is not publicly accessible. We, however, assume access to an existing auxiliary classifier which is a milder assumption as it exists for most use cases such as demographics (Serengil and Ozpinar, 2021). Furthermore, you can train an auxiliary classifier even on an imbalanced dataset as we show in this paper for the case of the CIFAR10 dataset.

Moreover, in situations with extremely imbalanced class distributions learning fair generative models is a difficult task as in these cases the original dataset may not contain sufficient samples of the underrepresented groups/classes to enable training a fair model. To solve this problem, we propose that any biased generative model can be debiased by training a new model of the same architecture (or alternatively retraining the same model), on a balanced dataset extracted from the existing model. The new, balanced dataset is generated from the unbalanced model by searching its latent space. We demonstrate that training on this balanced data results in a fairer model, with no loss to image quality.

To enable this procedure, we propose a novel fairness loss function, which is used to ensure that samples from the model have equal probability of belonging to each class in the data distribution. This loss function in addition to the proposed training methodology achieves state-of-the-art results in mitigating representation bias. We show results on the StyleGAN2 (Karras et al., 2020) architecture when trained on the flicker-faces high-quality (FFHQ) dataset (Karras et al., 2019) for generating racially diverse images. Our approach is generalizable to any GAN model trained on any biased dataset. We do not make any assumptions as to the architecture of the generative model or the nature of the dataset. To further investigate the generalizability of this approach to a larger class of problems, we apply the same approach to a class-imbalanced CIFAR10 dataset and show that we can learn the balanced data distribution almost as well as when training on a twice as large balanced dataset. In figure 1 we qualitatively show examples of images generated randomly using the standard StyleGAN2 model and the proposed approach. We visually observe a change in the number of people of color in the two images.

We make the following contributions in this paper:

- We propose a simple privacy-friendly approach to debias an existing generative model through model rebalancing without using the original training or reference dataset.

- We propose a novel fairness loss function that can be used together with a latent evolutionary search to mitigate biases in the existing GAN.

- We empirically show that this approach achieves significantly lower bias while maintaining the image quality.

- We highlight the potential of this approach to be applied to any GAN, trained for any task. We show results when applied to a class-imbalanced CIFAR10 dataset containing a long-tail class distribution.

- We discuss the shortcomings of the popularly used Frechet inception distance (FID) in measuring image quality in the case of class or demographic biases. We show that it is sensitive to the balance in the underlying data distribution and should not be used when studying biases if a balanced reference dataset is not available.

## 2 Methodology

In this section, we present our approach for mitigating biases in GANs using data generated from itself and a novel loss function. We start by formally formulating the problem in subsection 2.1, followed by a baseline approach in section 2.2. We discuss our proposed approach in the section 2.3, the loss function definition in section 2.3.3, and lastly the implementation details of the approach in section 2.5.

### 2.1 Problem Definition

Let's assume there exists some biased dataset with data distribution $p_{bias} : \mathcal{X}, \mathcal{D} \rightarrow \mathbb{R}$ over a set of classes $\{d_1, d_2, ..., d_n\} \in D$ and observed variables $x \in \mathcal{X}$. This dataset is used to train a generative model, parameterized by $\theta$, to learn the training data distribution. The generative model learns a distribution $p_\theta : \mathcal{X} \rightarrow \mathbb{R}$. Let us also consider an ideal fair data distribution referred to as $p_{ref} : \mathcal{X}, \mathcal{D} \rightarrow \mathbb{R}$. Now, the primary definition of fairness, especially in the case of demographic fairness is that, when randomly sampling images from a GAN, the expected number of samples for different classes is equal (Hutchinson and Mitchell, 2019). To mathematically state this, let's assume a Gaussian random latent vector $z \in \mathbb{R}^m$, generative model $G : \mathbb{R}^m \rightarrow \mathbb{R}^{n \times n}$ and an auxiliary classifier $\mathcal{C} : \mathbb{R}^{n \times n} \rightarrow \{1, 2, \ldots, |D|\}$ that classifies that data into the corresponding classes. We define an indicator random variable $\zeta$. This random variable is equal to 1 when the randomly sampled image from the generator $G$ corresponds to the target class, represented by a one-hot vector $d_i \in \mathbb{R}^{|D|}$, and 0 otherwise.

$$\zeta_{d_i}(z) = \mathbb{1}[C(G(z)) == d_i] \tag{1}$$

Now, we can define the fairness of a generative model with respect to each class as follows,

$$\mathbb{E}_{z \sim p_\theta}[\zeta_{d_i}(z)] \approx 1/|D| \tag{2}$$

## 2.2 Baseline: Importance Reweighting

As a baseline, we have implemented a modified version of the importance reweighting method proposed in (Choi et al., 2020; Grover et al., 2019). The authors had originally proposed an approach that utilizes a reference dataset, using which, they reweigh the data points in the biased dataset based on the ratio of densities assigned by the biased data distribution as compared to the reference data distribution. We derive a modified version of the proposed loss function with milder assumptions i.e. a reference dataset is not available but we have access to an auxiliary classifier $\mathcal{C}$. Originally the authors had defined the bias mitigation loss function as follows,

$$\begin{aligned}
\mathbb{E}_{x \sim p_{ref}}[l(x, \theta)] &= \mathbb{E}_{x \sim p_{bias}}\left[\frac{p_{ref}(x)}{p_{bias}(x)} l(x, \theta)\right] \\
&\approx \frac{1}{T} \sum_{i=1}^{T} w(x_i) l(x_i, \theta) := \mathcal{L}(\theta, \mathcal{D}_{bias}).
\end{aligned} \tag{3}$$

Given that both the reference and the biased datasets are randomly drawn from the same overall data distribution (assumption made in (Choi et al., 2020)), it implies that,

$$p_{ref}(x|d_i) = p_{bias}(x|d_i) \quad \forall d_i \in D. \tag{4}$$

Additionally, given an equally representative reference dataset, we can state that, $p_{ref}(d_i) = 1/|D| \forall d_i \in D$. We can thus, rewrite the loss definition as follows,

$$\begin{aligned}
\mathbb{E}_{x \sim p_{ref}}[l(x, \theta)] &= \mathbb{E}_{x \sim p_{bias}}\left[\frac{1/|D|}{p_{bias}(\mathcal{C}(x))} l(x, \theta)\right] \\
&\approx \frac{1}{T} \sum_{i=1}^{T} w(x_i) l(x_i, \theta) := \mathcal{L}(\theta, \mathcal{D}_{bias}).
\end{aligned} \tag{5}$$

where, $\mathcal{C}(x)$ is the class label of x as determined by an auxiliary classifier $\mathcal{C}$. Here, $w(x_i) := \frac{1/|D|}{p_{bias}(\mathcal{C}(x))} = \frac{1/|D|}{|C(x_i)|/|\mathcal{D}_{bias}|}$. For the entire proof please refer to the appendix A.

## 2.3 Proposed Approach

We assume a setting where there exists a trained generative model that is extremely biased wrt to the distribution of classes (e.g. protected attributes for face image generation). We assume we do not have any reference datasets available nor do we have access to the original training dataset.

Our approach consists of two parts. The first part is generating balanced data from the existing generative models using a latent space search algorithm. Secondly, we use the generated data in conjunction with a novel bias mitigation loss function to train the same generative model.

### 2.3.1 Data Generation

The StyleGAN model, like most GANs, has disentangled latent spaces. This implies that there exist directions or subspaces in the latent space corresponding to different attributes or classes of the dataset. We can make

use of these disentangled subspaces to generate balanced synthetic datasets even from extremely biased generative models. To do so, we employ an algorithm similar to breadth-first search in latent space (Jain et al. 2023).

The algorithm consists of two parts. The first one is finding a starting vector to begin the search and the second is the search itself. We find a starting vector through random rejection sampling using an auxiliary classifier for feedback. We then sample around the starting point in an iterative manner such that we are always moving outwards. This algorithm can also be seem as an evolutionary algorithm. This approach allows us to generate demographically balanced data in a zero-shot setting. The search algorithm is summarized in Algorithm 2

### 2.3.2 Training

We use this generated data for rebalancing the same generative model by training on it. Given the balanced nature of the generated dataset and the ability of generative models to generate highly realistic images, we can expect to learn the reference data distribution. Let us assume that the generated dataset has a data distribution $p_{G_\theta} : \mathcal{X}, \mathcal{D} \to \mathbb{R}$. Now, given the ability of GANs to generate highly realistic images, $p_{G_\theta}(x|d) \approx p_{ref}(x|d) \forall d \in D$. Also, since we generate a balanced synthetic dataset $p_{G_\theta}(d) = p_{ref}(d) \forall d \in D$. This implies that, $p_{G_\theta}(x) \approx p_{ref}(x)$. Thus, we finally train a generative model with distribution $p_{G'_\theta}$ that approximately tries to learn the data distribution of a perfectly balanced set.

However, in practice, this may not always be the case. Generative models are well known to favor the generation of classes and images that the discriminator finds difficult to classify (i.e. mode collapse). Thus, we further propose a loss function that specifically focuses on generating a balanced class distribution.

### 2.3.3 Fairness Loss Function

Choi et al. (2020) reweighted samples based on a reference dataset distribution, however, this does not guarantee learning a balanced distribution. Without assuming access to a reference dataset, the proposed loss function tries to directly minimize the deviation from an ideal class distribution. We do so by minimizing the expected number of samples belonging to each class $d_i$ from the ideal value $1/|D|$. We take a weighted sum of the deviation for each demographic subgroup using the weights $\lambda_{d_i}$ which is inversely proportional to the current bias of the generative model. We define it similar to the density ratio estimation weights i.e. $\lambda_{d_i} = \frac{1/|D|}{p_{bias}(d_i)}$ This gives more importance to the loss corresponding to the underrepresented subgroups.

$$\mathcal{L}(\theta) = \mathcal{L}_{stylegan} + \sum_{d_i \in D} \lambda_{d_i} max(0, 1/|D| - \mathbb{E}_{z \sim p_\theta}[\zeta_{d_i}]) \tag{6}$$

The expectation is calculated via Monte Carlo averaging over the batch of images as shown in equation 7. In practice, this loss function is implemented as shown in Algorithm 1.

$$\mathbb{E}_{z \sim p_\theta}[\zeta_{d_i}(z)] = \frac{1}{|B|} \sum_{z \in B} \mathbb{1}[\mathcal{C}(\mathcal{G}(z)) == d_i] \tag{7}$$

### 2.4 Evaluation Metrics

**Fairness Measure**   To evaluate the fairness of the generative model for different classes or demographic subgroups, we utilize a fairness discrepancy metric. The metric was first proposed by (Choi et al. 2020) to measure the discrepancy between the expected marginal likelihoods of $d$ as per $p_{ref}$ and $p_\theta$. It can be mathematically formulated as follows,

$$f(p_{ref}, p_\theta) = |E_{x \sim p_{ref}}[p(d|x)] - E_{z \sim p_\theta}[p(d|z)]|_2 \tag{8}$$

---

**Algorithm 1** Fairness Loss Function Computation

---

**Input:** A generative model $G$, an auxiliary ethnicity classifier $C$, mini-batch size $B$, weighting for each demographic subgroup $\lambda_d$, number of demographic subgroups $|D|$

**Output:** Loss value.

1: $v_l \leftarrow \text{random}(B)$ {Initialize $B$ latent vectors}
2: $\mathcal{I}_B \leftarrow G(v_l)$
3: $C\_logits \leftarrow \arg\max \mathcal{C}(\mathcal{I}_B)$
4: $C\_one\_hot \leftarrow \text{one\_hot\_encode}(C\_logits, \dim = |D|)$
5: $C\_count \leftarrow \text{sum}(C\_one\_hot, \text{axis} = 0)$
6: $E\_\zeta_d \leftarrow 1/|B| \times C\_count$
7: $E\_\zeta_d \leftarrow \text{nn.ReLU}(1/|D| - E\_\zeta_d)$
8: $\text{loss} \leftarrow \lambda_d \times E\_\zeta_d$
9: **return** loss

---

**Algorithm 2** Algorithm for Generating Synthetic Data

---

**Input:** A generative model $\mathcal{G}$, an auxiliary classifier $\mathcal{C}$, target race $t$, starting latent vector found using random sampling $v_s$ such that $\mathcal{C}(\mathcal{G}(v_s)) = t$, number of mutations $n$, max range of random mutation $\delta$, max number of iterations for a particular starting vector $\xi$

**Output:** A list of latent space vectors *out*.

1: $queue \leftarrow []$ {Initialize empty list}
2: $out \leftarrow []$
3: $iter \leftarrow 0$
4: $queue.\text{enqueue}(v_s)$
5: **while** $\text{len}(queue) \neq 0$ and $iter \leq \xi$ **do**
6:     $v_c \leftarrow \text{queue.pop}()$
7:     $\mathcal{I} \leftarrow \mathcal{G}(v_c)$
8:     **if** $\mathcal{C}(\mathcal{I}) == t$ **then**
9:         $out.\text{append}(v_c)$
10:         $iter \leftarrow iter + 1$
11:         **for** $j = 0$ **to** $n$ **do**
12:             $v_j \leftarrow v_c + \text{random}(\text{range} = [-\delta, \delta])$
13:             **if** $\text{dist}(v_j, v_s) > \text{dist}(v_c, v_s)$ **then**
14:                 $queue.\text{enqueue}(v_j)$
15:             **end if**
16:         **end for**
17:     **end if**
18: **end while**
19: **return** List of latent vectors corresponding to the target class - *out*.

---

Like before, we eliminate the reference dataset in the metric by stating that all classes should be equally likely to occur. This simplifies the way the fairness of the GAN model is calculated while also negating the need for a real reference dataset. Equation 9 shows the updated fairness metric after incorporating this.

$$f(p_{ref}, p_\theta) = |1/|D| - E_{z \sim p_\theta}[p(d|z)]|_2 \tag{9}$$

We compute the metric using Monte Carlo averaging. In our experiments, we have averaged this for 5 different evaluation sets consisting of 10,000 randomly drawn samples.

**Sample Quality Measure** The Frechet inception distance (FID) (Heusel et al., 2017) is used to measure the similarity between two data distributions. Indicatively, FID is calculated using the distance between the 2048-dimensional activations of sets of images of the pool-3 layer of the inception network. This is popularly used in measuring the quality of images generated by a GAN model. However, it is very sensitive to the

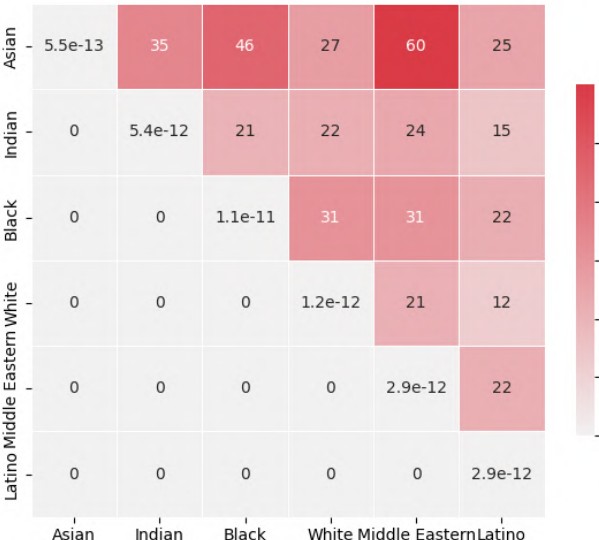

Figure 3: Heatmap showing the FID scores between different racial groups present in the FFHQ dataset. The high scores in the cross-racial distributions show that the FID score is sensitive to the demographic distribution of the underlying datasets.

reference dataset distribution (Borji, 2022). This makes the metric inadequate in bias-mitigation scenarios where a representative real dataset may not be present, as in our case. We observed that taking the FID score between sets of real facial images from the FFHQ dataset is quite high when two sets are from different ethnic groups. These images are collected in similar environments and have similar image quality, however, the FID score is in the range of 12-60. This indicates that the metric is not just capturing the image quality; if this was the case the value would have been significantly lower. For reference, the FID score between two disjoint sets of images belonging to the White population is 1.76. We see a similar trend with the Kernel Inception Distance (KID) as discussed in the appendix B.

Thus, we argue that in the case of bias mitigation, the Inception network-based metrics that require a reference dataset are not a good measure of image quality when a balanced reference dataset is not present. To further emphasize this point, we have also taken another use-case of generative modeling for another class imbalance problem in section 3.2 wherein a reference dataset is present. Please refer to section 3 for further analysis of the experimental results. However, we still report results on the metrics used by the original StyleGAN2 authors (Karras et al., 2020) including FID, KID, inception score (IS), precision, and recall along with the perceptual path length (PPL) metric.

Given the issues with reference-based image quality metrics, we also report results for evaluating the image quality based on other no-reference image quality assessment metrics - Blind/Referenceless Image Spatial Quality Evaluator (BRISQUE) (Mittal et al., 2012a), Naturalness Image Quality Evaluator (NIQE) (Mittal et al., 2012b) and Clip-IQA (Wang et al., 2023). Additionally, we also use the referenceless image-quality metric EQ-Face (Liu and Tan, 2021) that is specifically used for facial images.

## 2.5 Datasets and Implementation Details

To show the efficacy of the proposed approach towards introducing demographic fairness we show results on generating racially fair data using the FFHQ dataset. The dataset contains extreme biases in terms of the racial groups, wherein 69% of the images are of Caucasians and only about 4% are of Africans (Maluleke et al., 2022). We train our own StyleGAN2 model on 4 Nvidia RTX 8000 GPUs using their publicly available codebase to maintain consistency across all the results. The model was trained at a resolution of 256x256 for

Table 1: Table comparing different approaches proposed with the baselines on the fairness metric for racial parity in the generated images.

| Dataset/ Method | Fairness ↓ |
|---|---|
| FFHQ (reweighing) | $0.4588 \pm 0.0052$ |
| FFHQ | $0.4525 \pm 0.0064$ |
| + bias loss | $0.4503 \pm 0.0063$ |
| + Syn data | $0.1024 \pm 0.0053$ |
| + freezeD=5 | $0.1056 \pm 0.0044$ |
| Syn data | $0.0992 \pm 0.0041$ |
| + bias loss | $\mathbf{0.0947 \pm 0.0071}$ |

25 million iterations using the config-f parameters for both training and finetuning. We use a pre-trained ethnicity classifier from the DeepFace (Serengil and Ozpinar, 2021) to serve as an auxiliary classifier.

We generated 50,000 unique synthetic identities for 6 different racial groups - Blacks, Indians, Asians, Whites, Hispanic Latinos, and Middle Easterners. We also included controlled variations in pose, expression, and illumination to generate a total of 13.5 million images.

We also apply the proposed approach to the CIFAR10 dataset, by creating an imbalanced version of the dataset following (Shu et al., 2019; Yuan et al., 2023; Cao et al., 2019) such that it has a long-tailed class distribution. We use an imbalance ratio $\beta = \frac{N_{min}}{N_{max}}$ of 0.1; here $N_{max}$ is the most frequent class and $N_{min}$ is the least frequent class. Using this dataset we train a biased version of the StyleGAN model. The model was trained in a similar fashion using the cifar-config parameters to generate 32x32-sized images. Using the biased model, we generate 50,000 synthetic images per class; totaling half a million images. We trained a ResNet32 model on the same imbalanced dataset to serve as an auxiliary classifier using (Du et al., 2023). This model achieved a balanced accuracy of 94% on the CIFAR10 test set.

We train the following variants of the StyleGAN model and have used these abbreviations in the tables and corresponding discussions. These serve as ablations for the final proposed model configurations.

- *reweighting*: Baseline approach using the importance reweighting technique (Choi et al., 2020).

- *+ bias loss*: Adding the bias mitigation loss to the training procedure. We weigh the StyleGAN loss and the bias loss equally.

- *+ Syn data*: Finetuning the existing model trained on the original dataset on the generated synthetic dataset. The finetuning step is always done using the proposed bias mitigation loss function with a 10 times lower learning rate to preserve elements of the learned generative model.

- *+ freezeD=5*: Freezing the last five layers of the original discriminator model and then retraining on the synthetic data. This is done to preserve the image quality when finetuning on the generated dataset.

Overall, we propose training a new model with the same architecture on the synthetic data with the bias loss (last row in the results tables). This is not only more privacy-aware as compared to retraining the same model but also does not assume access to the biased GAN discriminator. We delve deeper into the results obtained using each of these approaches in terms of fairness and image quality in the next section.

## 3 Experimental Results

In this section, we empirically validate the proposed approach of learning fairer generative models. We structured the results section to answer the following two questions,

- Can we achieve a fairer generative model that is equally representative wrt to the different classes in the dataset?

Table 2: Comparison of the baseline model with the proposed approach on the no-reference image quality metrics - ED-Face, NIQE, Brisque, and Clip-IQA when training the model for generating racially diverse faces.

| Dataset/ Method | EQ-Face ↑ | NIQE ↓ | Brisque ↓ | CLIP-IQA ↑ |
|---|---|---|---|---|
| FFHQ (reweighing) | 0.5491 ± 0.1507 | 4.8018 ± 1.0926 | 7.0015 ± 8.2990 | 0.6814 ± 0.0940 |
| FFHQ | 0.5496 ± 0.1476 | 4.9053 ± 1.2079 | 6.9652 ± 8.3043 | 0.6835 ± 0.0905 |
| + bias loss | 0.5396 ± 0.1483 | **4.7568 ± 1.0676** | **6.3721 ± 8.1770** | 0.6802 ± 0.0903 |
| + Syn data | **0.5820 ± 0.1247** | 4.8395 ± 1.0846 | 7.5867 ± 7.5897 | 0.6877 ± 0.0797 |
| + freezeD=5 | 0.5815 ± 0.1250 | 4.8507 ± 1.0328 | 7.9044 ± 7.6993 | **0.6936 ± 0.0785** |
| Syn data | 0.5757 ± 0.1294 | 4.9408 ± 1.1059 | 8.7538 ± 7.6573 | 0.6803 ± 0.0811 |
| + bias loss | 0.5739 ± 0.1255 | 4.7675 ± 0.9914 | 8.1593 ± 7.4206 | 0.6603 ± 0.0807 |

Table 3: Experimental results showing the comparison of the baselines against the proposed approach across different metrics discussed in section 2.4 for generation of racially diverse faces. The scores are measured wrt to the racially imbalanced FFHQ dataset due to the lack of a relevant balanced dataset.

| Dataset/ Method | FID ↓ | KID ↓ | Precision ↑ | Recall ↑ | IS ↑ | PPL ↓ |
|---|---|---|---|---|---|---|
| FFHQ (reweighing) | 5.11 | 0.0015 | 0.6665 | 0.4353 | **5.28 ± 0.07** | 135.73 |
| FFHQ | **4.60** | **0.0014** | 0.6842 | **0.4299** | 5.14 ± 0.06 | 180.44 |
| + bias loss | 5.55 | 0.0021 | 0.7128 | 0.3866 | 5.04 ± 0.05 | 259.44 |
| + Syn data | 23.81 | 0.0168 | 0.8152 | 0.1679 | 3.66 ± 0.04 | 182.10 |
| + freezeD=5 | 24.15 | 0.0169 | 0.8159 | 0.1592 | 3.67 ± 0.02 | 184.36 |
| Syn data | 24.53 | 0.0174 | 0.8037 | 0.1484 | 3.79 ± 0.03 | **88.46** |
| + bias loss | 27.17 | 0.0186 | **0.8245** | 0.1012 | 3.54 ± 0.04 | 110.76 |

- Can we do so without degrading the quality of generated images?

We report results on the FFHQ dataset in section 3.1 and on the imbalanced CIFAR10 dataset in section 3.2

## 3.1 Results on the FFHQ Dataset

**Results on Fairness.** We have reported the results pertaining to the demographic disparity in generative models in table 1. We see the model trained on the synthetically generated balanced image set with the bias loss (last row in the table), improves the fairness of the standard StyleGAN2 trained on the FFHQ dataset by almost 5 times.

We also see that the proposed bias loss plays an important role as well when comparing the StyleGAN2 model trained on the FFHQ dataset (row 2) versus finetuning the model on the FFHQ dataset with the bias loss (row 3). Further, this is lower than when using the baseline importance reweighting technique (row 1) which has a fairness metric value of 0.4588 in comparison to 0.4503 of the former.

**Results on Image Quality.** We summarized the results on image quality using referenceless IQA metrics in table 2 and on other metrics employed by the original StyleGAN2 authors in table 3. When comparing the referenceless image qualities for the proposed approach with the baseline StyleGAN2 model trained on the FFHQ dataset, we see that the model trained with the *Syn data + bias loss* configuration has comparable image quality scores. The EQ-Face metric in particular has been trained on facial images to judge their quality for performing facial recognition. We in fact see higher EQ-face image quality scores when using our proposed approach with a score of 0.5739 as compared to the traditional model with a score of 0.5496. The baseline traditional StyleGAN2 model has a better BRISQUE score of 6.96 as compared to the proposed approach which has a score of 8.15. The NIQE and CLIP-IQA scores are similar across all the models.

As discussed in section 2.4, the traditional metrics utilized by the authors of the StyleGAN2 paper are partially irrelevant to a class imbalance problem. We still report results on these metrics in table 3

Table 4: Comparison of the proposed approach and the baseline trained on a long tail CIFAR10 dataset ($\beta$=0.1) with the model trained on the balanced dataset. The metrics are measured wrt to the original balanced CIFAR10 dataset. *The proposed approach performs almost as well as training on balanced data (row 5-8 vs row 1) with 40% of the training data and that too imbalanced.*

| Dataset/ Method | FID ↓ | KID ↓ | Precision ↑ | Recall ↑ | IS ↑ | PPL ↓ |
|---|---|---|---|---|---|---|
| Balanced data | **4.36** | 0.00155 | 0.6330 | 0.5684 | **9.49 ± 0.10** | **18.90** |
| Imbalanced data (reweighting) | 11.28 | 0.00614 | 0.6429 | **0.5100** | 8.33 ± 0.08 | *19.38* |
| Imbalanced data | 11.13 | 0.00628 | 0.6937 | 0.4398 | 8.30 ± 0.12 | 20.66 |
| + bias loss | 10.30 | 0.00568 | 0.6837 | 0.4632 | 8.47 ± 0.17 | 22.85 |
| + Syn data | 6.44 | 0.00201 | 0.6974 | 0.3860 | 9.19 ± 0.08 | 23.09 |
| + freezeD=5 | 6.56 | 0.00213 | **0.6984** | 0.3946 | 9.21 ± 0.10 | 23.19 |
| Syn data | 6.38 | 0.00152 | 0.6771 | 0.3905 | 9.44 ± 0.10 | 23.28 |
| + bias loss | *6.32* | **0.00146** | 0.6800 | 0.3877 | **9.49 ± 0.11** | 23.96 |

Table 5: Table showing the comparison of various methods measuring the deviation of the probability of generated samples belonging to each class of the CIFAR10 dataset from being equiprobable (i.e. equal to 0.1).

| Dataset/ Method | Fairness ↓ |
|---|---|
| Balanced data | 0.0286 ± 0.0018 |
| Imbalanced data (reweighting) | 0.2313 ± 0.0023 |
| Imbalanced data | 0.2279 ± 0.0020 |
| + bias loss | 0.2183 ± 0.0029 |
| + Syn data | 0.0290 ± 0.0019 |
| + freezeD=5 | 0.0285 ± 0.0015 |
| Syn data | 0.0288 ± 0.0022 |
| + bias loss | **0.0281 ± 0.0033** |

## 3.2 Results on the CIFAR10 Dataset

To further validate our approach of mitigating biases in generative models wherein there is a high imbalance in the classes, we have applied the same approach to the CIFAR10-imbalanced dataset with a long tail distribution. We have some interesting observations based on these results, especially since in this case we have a balanced reference set and can accurately compute metrics such as FID and KID. Thus, we have computed all the reference-based metrics with respect to the original balanced CIFAR10 dataset. By doing so we observed that the FID and KID scores in fact improve when we just apply the bias loss. As seen in Table 4 the FID score improves from 11.28 to 10.30. This clearly shows the bias loss pushes the generative model to generate a more balanced dataset that better resembles the ideal underlying distribution. This also acts as a further validation that the FID is highly sensitive to the class distribution and is lower when the two distributions have similar ratios of the classes, going away from the popular belief that it only measures the image quality.

The proposed training regime achieves FID and fairness scores of 6.32 and 0.0281 which are almost similar to when trained on the fully balanced CIFAR10 dataset which yields scores of 4.36 and 0.0286 respectively. This is also 10 times lower than the fairness value when trained using the importance weighting technique and directly on the imbalanced dataset. It is also important to note that the imbalanced CIFAR10 dataset only has 20,431 samples in comparison to the original CIFAR10 dataset which contains 50,000 samples. *Thus, the proposed approach despite having less than half the amount of original data is able to learn the true balanced distribution almost as well as when training on the full balanced dataset.*

We also looked at how the generated images performed on the different no-reference image quality assessment metrics. We report results on the BRISQUE and the CLIP-IQA metrics. However, given the extremely small size of images in the CIFAR10 dataset, we were not able to compute the NIQE scores. But in reference to the BRISQUE and Clip-IQA, the model trained with the bias loss performs at par with the model trained on

Table 6: Comparison of the proposed approach applied to the imbalanced CIFAR10 dataset on no-reference image quality metrics.

| Dataset/ Method | Brisque ↓ | CLIP-IQA ↑ |
|---|---|---|
| Balanced data | **60.13 ± 35.18** | 0.51 ± 0.05 |
| Imbalanced data (reweighting) | 62.57 ± 35.02 | 0.51 ± 0.05 |
| Imbalanced data | 62.14 ± 34.57 | 0.51 ± 0.05 |
| + bias loss | *60.87 ± 34.13* | 0.51 ± 0.05 |
| + Syn data | 61.51 ± 35.77 | 0.51 ± 0.05 |
| + freezeD=5 | 61.33 ± 35.84 | 0.51 ± 0.05 |
| Syn data | 62.21 ± 36.13 | 0.51 ± 0.06 |
| + bias loss | 62.43 ± 36.34 | 0.51 ± 0.06 |

the balanced set, thus showing that it improves significantly on fairness without compromising on the image quality.

## 4 Related Work

Various researchers have studied fairness in generative models and have pointed out the extremely biased nature of popularly used existing generative models including the StyleGAN models trained on the FFHQ dataset (Maluleke et al., 2022; Tan et al., 2020). To tackle this issue researchers have worked on improving the fairness in these models. Choi et al. (2020) presented an approach to train fair generative models by weighting the loss based on the fairness importance of the sample. They use an unlabelled reference dataset for learning the desired distribution using a density ratio technique. In a similar set-up with a smaller fair reference dataset (Teo et al., 2023) proposed an approach to tune a biased model on the fair dataset. In this work, we simplify the "fairness" meaning by simply stating that the ideal distribution is where the expectation of every class is equal. This negates the need for a reference dataset. Kenfack et al. (2022) studied fairness in situations when the dataset is fair, yet there are biases in the generative model. They proposed an approach of group-wise gradient clipping in the discriminator to ensure that the generator doesn't favor a particular class. Early work in the field that focused on fairness in generative models focused on using labeled datasets or creating labeled datasets (Xu et al., 2018; Sattigeri et al., 2018). This allows them to learn a downstream classifier for target attributes. In this work, we do not assume that a demographically labeled or unbiased dataset exists. In fact, the focus of this work is on making use of existing models and classifiers to mitigate biases in them without using any real data. Kenfack et al. (2021) proposed using conditional GANs or an ensemble of GAN models to improve fairness in situations where GAN models favor particular classes. Jalal et al. (2021) proposed a new formulation of fairness for image-to-image translation models called Conditional Proportional Representation and have argued that Representation Demographic Parity is incompatible in this setting. This work is not directly comparable to our work since the notion of CPR is not extendable to GANs. Karakas et al. (2022) proposed an approach to introduce fairness in StyleGAN by directly modifying style channels that control particular attributes. They showed results on gender and other attributes such as eyeglasses.

Interestingly, there has been a wider focus on generating more diverse images in terms of protected or un-protected facial attributes from biased generative models (Jain et al., 2023; Tan et al., 2020; Colbois et al., 2021; Parmar et al., 2022; Liu et al., 2019; He et al., 2019; Shen et al., 2020; Dabouei et al., 2020). We believe this trend in the research community arises from the lack of publicly available datasets used to train large models along with the computational cost associated with the training of such models. In this work, we have tried to address both of these issues.

Researchers have also shown interest in proposing other fairness measuring metrics for generative models that take into account the inaccuracy of the auxiliary classifier (Teo et al., Teo and Cheung, 2021).

The generation of synthetic datasets is easier than the collection of real datasets and also preserves privacy. Researchers have shown its applicability in various tasks such as multi-view representation learning (Jahanian

et al., 2021), face-swap detection (Jain et al., 2022), face morphing attack detection (Ivanovska et al., 2022) and even in finance (Assefa et al., 2020). Thus it is important that these generative models are unbiased so that the same biases are not propagated in the downstream tasks. Another line of related research explores the generation of images for data augmentation in imbalanced situations Mullick et al. (2019); Dablain et al. (2022).

A closely related topic of generative modelling for long tail distributions has been studied by Rangwani et al. (2022, 2023). However, these works study training procedures where the labels are known i.e. you can train a conditional generative model which is not the task we have addressed in this paper. Models generating facial images are generally trained in an unconditional manner wherein class information cannot directly be utilized.

Recently, Shumailov et al. (2023) showed that recursively training on generated data leads to model collapse. The model begins losing information from the tails of the distribution or entangles multiple modes of the original distribution. In this work, we only do one iteration of retraining the generative model on synthetic data while specifically focusing on generating diverse images, thus, minimizing the risk of model collapse.

## 5 Conclusion

In conclusion, we present a novel approach to building a fair GAN from an existing biased GAN. We have specifically proposed a privacy-aware approach that is useful while handling sensitive biometric data such that we do not use any of the original training data. This is also important in cases where access to the original training data is no longer available. Further, we eliminate the need for a balanced reference dataset. We also propose a novel bias mitigation loss function. We experimentally validate our approach on two different tasks of racial bias in facial image generation and imbalanced-CIFAR10 image generation. We see significant improvements in the fairness metrics while maintaining the image quality. In fact on the CIFAR10 dataset, we show that we are able to closely learn the true balanced distribution even on the imbalanced dataset which contains less than half as many samples. We believe this approach is generalizable to any generative model trained for any task. In the future, it would be interesting to see if such approaches can also be applied to diffusion-based generative models.

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
