# OpenReview forum: "Fair GANs through model rebalancing for extremely imbalanced class distributions"
_TMLR — Rejected by TMLR_

### Review · Reviewer_TDE4 · 2024-05-30

**Summary Of Contributions:**

The paper proposes a method to balance GAN, and in particular StyleGAN, generation by starting from an unbalanced GAN and a trained classifier. The balance is achieved by, on one hand, generating balanced data using an evolutionary algorithm, and then training a new generative model on this data, and on the other hand by introducing a new fairness loss to distribute the synthetic samples similarly to an uniform distribution. The proposed method is tested in two scenarios: mitigating the ethnic bias in facial image generation and on an unbalanced version of CIFAR-10. The approach can be used on any unbalanced GAN as it does not require access to any external data.

**Audience:**

Yes

**Broader Impact Concerns:**

The paper deals with the generation of faces of different ethnicities, which potentially might lead to the misuse of such generated content. A discussion of the impact of this technology is needed.

**Claims And Evidence:**

Yes

**Requested Changes:**

- **Discussion of the literature on long-tail generation:** A discussion of works similar to (Rangwani 2022) is needed as they are dealing with a different, but related problem, that is learning a balanced generative model by having access to an unbalanced training distribution. In particular what is missing is showing that the proposed approach works in extremely imbalanced class distribution, such as CIFAR10 with $\beta={0.01,0.001}$.
- **Quantify the model coverage loss:** By providing a measure of recall and precision of the re-trained model on only synthetic data in extremely imbalanced scenarios, it would strengthen the point that the final model can be used in settings where balancing is needed while retaining generative power.

**Strengths And Weaknesses:**

## Strengths

The task the authors study is an important one, i.e. understanding and mitigating the bias in generative models, tools that have become ubiquitous and generally available. Being able to mitigate such behavior would lead to more trustworthy AI. The study focuses on StyleGAN, but to me this does not weaken the claims of the authors. The text is clear and well-presented, and it flows nicely. Lastly, the proposed approach is sound, especially the data generation part, which greatly exploits the latent space traversal which is possible only on disentangled latent spaces, such as that of StyleGAN.
## Weaknesses
1. **Extreme imbalance:** The authors mention extremely imbalanced class distributions in the title, while from the experimental section, the only explicit mention of imbalance is the CIFAR10 dataset with the $\beta$ ratio of 0.1. In the long-tail generation research, it is common to see $\beta$ ratios of 0.01 and 0.001, e.g. in (Rangwani 2022, and Rangwani 2023), which represent extremely unbalanced distribution. A discussion of those works, and experiments in the real imbalanced scenario are missing.
2. **Role of bias loss:** From the quantitative evaluation, it looks like the role of the proposed fairness loss is way minor compared to the training with the synthetic data. For example, in Table 1, adding the bias loss leads to a reduction of the fairness metric of less than 1%, while adding the synthetic data brings a reduction of around 77%. Does the bias loss bring to qualitatively different results? A discussion and (possibly) a qualitative comparison of samples with and without bias loss might strengthen the claim.
3. **Predetermined biases:** One drawback of the proposed approach is that it can mitigate only the bias predetermined by the external classifier. While it does weaken my evaluation of the paper, being limited by the classifier attributes seems a big limiting factor.
4. **Evaluation:** While the authors (correctly) criticize metrics such as FID to evaluate the task at hand, no alternative is presented to better quantify the goodness of the proposed method. I would suggest an approach similar to that of (Stein 2023) which could mitigate the bias present in the Inception network used for FID computation.
5. **Generative Collapse:** While the authors discuss the line of research regarding the iterative collapse of generative models in Section 4, the quantitative evaluation of the coverage of the space with the fully-synthetic trained model does not include extremely imbalanced scenarios, which could greatly damage the coverage. Although I am inclined to believe one iteration is not enough to make the model collapse, the training is performed on synthetic data coming from a GAN, which is already itself prone to collapse.
6. **Typos and missing definition:** There are some typos in the text, e.g. the broken link to the first Figure at the beginning of page 3, or the missing definition of $D_{bias}$ in Equation 3.

---

Stein, George, et al. "Exposing flaws of generative model evaluation metrics and their unfair treatment of diffusion models." _NeurIPS_ (2023).

Rangwani, Harsh, et al. "Improving gans for long-tailed data through group spectral regularization." _ECCV_. 2022.

Rangwani, Harsh, et al. "Noisytwins: Class-consistent and diverse image generation through stylegans." _CVPR_. 2023.

---

> ### Author Response · Authors · 2024-07-17
> **Response to Reviewer TDE4**
>
> We sincerely thank Reviewer TDE4 for their constructive comments and valuable feedback on our submission. We are grateful that the reviewer acknowledges the importance of the problem our paper addresses, and appreciates the clarity and presentation of the text,  and the correctness of our proposed approach.
> Inspired by their feedback, we have run several new experiments and updated our draft accordingly. We respond to each of their comments below:
>
> $\newline$
>
> ### 1. Extreme imbalance
>
> Thank you for bringing to our attention these interesting works. In our opinion, this scenario is less a matter of learning a fair generative model, and more a matter of learning from a few samples as tackled by Rangwani et al. 2022 and 2023 [1,2]. We focus on scenarios where there is sufficient data to train the model, but it is imbalanced. We will make this clear in the paper.
> Nevertheless, we agree with the reviewer that the works are related and relevant and we have thus included a discussion on these papers. We have also performed additional experiments in these settings and have included results in the Appendix in section C.
>
> $\newline$
>
> ### 2. Role of bias loss:
>
> Your observation is correct. Our main proposed approach is based on generation and usage of balanced synthetic datasets for introducing fairness. The bias loss is in addition to the said approach and has shown small but significant improvements in both in the fairness metric and FID scores for CIFAR-10 LT generation. Since previous approaches such as Choi et al. [3] were loss based, this acts as an alternative to such approaches and has consistently shown better results. Finally, in extremely imbalanced situations such as $\beta$=0.001, generation of balanced synthetic datasets for the last few most underrepresented classes is not viable as shown in the Appendix C, here the bias loss becomes even more important.
>
> $\newline$
>
> ### 3. Predetermined biases:
>
> It is true that we must know which kind of bias we are trying to correct for. This may be the case for any effective bias mitigation system (although this is not a claim in the paper; we have yet to see a system that can correct for biases that are not known directly or indirectly). In comparison to previous works wherein you need a curated balanced reference dataset (**which should also be from the same data distribution as the training data for density ratio estimation**), this is a milder assumption. Creation or acquisition of such a balanced dataset is a significantly harder task and would also require defining the biases that need to be removed. For example with facial image datasets, to the best of our knowledge no such balanced dataset exists wherein the balancing criterions weren’t already predefined. Datasets such as RFW [4], BFW [5] define the racial groups that they want to balance. We would also like to point out that for density ratio estimation to work, you need a CIFAR like dataset where the distributional similarity between the train and test (small reference) sets are very high, however this is not the case for more real world applications such as demographic biases. Choi et al. [3] have predefined biases and then taken subsets of the **same** CelebA dataset to show results on demographic biases which is not a realistic scenario, as has Teo et al [6] for transfer learning.
>
> $\newline$
>
> ### 4. Evaluation:
>
> We have evaluated the goodness of other encoders including self supervised models such as SwAV and have seen similar trends wherein the biases in the training dataset negatively impact the Frechet distance scores. Thus, we opted to showcase the results on the CIFAR-10 LT datasets wherein a reference dataset for computing FID and other such metrics is available. We have however provided a reference to the mentioned paper in our article as it touches upon similar issues with popularly used metrics.
>
> $\newline$
>
> ### 5. Generative Collapse:
>
> That’s an interesting point and one that we have carefully considered. We have evaluated the model using multiple metrics on image quality and have visually inspected the images as well and have not noticed any signs of model collapse as shown by previous authors.
>
> $\newline$
>
> ### 6. Typos and missing definition:
>
> We thank the reviewer for pointing these out. We have made the relevant changes to the paper.
>
> $\newline$
>
> [1] Rangwani et al. "Improving gans for long-tailed data through group spectral regularization." ECCV. 2022.
>
> [2] Rangwani et al. "Noisytwins: Class-consistent and diverse image generation through stylegans." CVPR. 2023.
>
> [3] Choi et al. "Fair Generative Modeling via Weak Supervision" ICML 2020
>
> [4] Wang et al. "Racial faces in the wild: Reducing racial bias by information maximization adaptation network." ICCV 2019.
>
> [5] Robinson et al. "Face recognition: too bias, or not too bias?." CVPR-W 2020.
>
> [6] Teo et al. "Fair Generative Models via Transfer Learning" AAAI 2023.

---

> > ### Author Response · Authors · 2024-07-18
> > **Response to Reviewer TDE4 (continued)**
> >
> > Requested Changes:
> > 1. Discussion of the literature on long-tail generation:
> >
> > We have included results on long tail generation in Appendix section C for both $\beta$=0.01 and $\beta$=0.001. We have also provided a discussion on the papers by Rangwani et al. 2022, 2023 in the Related Works section 4. The tables below report a condensed version of the results for imbalance ratios of 0.01 and 0.001 respectively. Please see appendix C for results on image quality. For imbalance ratio of 0.01 we saw that the FID score improved from 23.70 to 14.80 with our approach with little to no degradation in the image quality. While in the case of imbalance ratio of 0.001, due to the imbalanced in both the GAN and the classification model, generating a balanced dataset proved to be difficult. We have discussed this in further detail in the paper.
> >
> >
> > | Dataset/ Method               | FID $\downarrow$ | KID $\downarrow$ | Precision $\uparrow$ | Recall $\uparrow$ | IS $\uparrow$   | PPL $\downarrow$ | Fairness Metric $\downarrow$   |
> > |-------------------------------|------------------|------------------|----------------------|-------------------|-----------------|------------------|------------------|
> > | Imbalanced Data (Reweighting) | 24.73            | 0.0133           | 0.6724               | 0.4370            | 7.13 $\pm$ 0.09 | 21.64            |  0.4032 $\pm$ 0.0014 |
> > | Imbalanced Data               | 23.70            | 0.0124           | 0.6879               | 0.4040            | 7.21 $\pm$ 0.07 | 21.39            | 0.4077 $\pm$ 0.0032 |
> > | + Bias Loss                   | 23.68            | 0.0125           | 0.6926               | 0.3903            | 7.17 $\pm$ 0.12 | 21.83            |  0.4016 $\pm$ 0.0039 |
> > | + Syn Data                    | 14.71            | 0.0043           | 0.6786               | 0.3357            | 9.00 $\pm$ 0.08 | 32.28            | 0.1713 $\pm$ 0.0016 |
> > | + FreezeD=5                   | 15.05            | 0.0046           | 0.6738               | 0.3463            | 9.00 $\pm$ 0.10 | 33.19            | 0.1722 $\pm$ 0.0023 |
> > | Syn Data                      | 14.86            | 0.0041           | 0.6666               | 0.3379            | 9.05$\pm$ 0.06  | 26.15            | 0.1721 +- 0.0016 |
> > | + Bias Loss                   | 14.80            | 0.0041           | 0.6627               | 0.3533            | 9.09 $\pm$ 0.08 | 26.05            | 0.1699 $\pm$ 0.0016 |
> >
> >
> >
> > | Dataset/ Method               | FID $\downarrow$ | KID $\downarrow$ | Precision $\uparrow$ | Recall $\uparrow$ | IS $\uparrow$   | PPL $\downarrow$ | Fairness Metric $\downarrow$   |
> > |-------------------------------|------------------|------------------|----------------------|-------------------|-----------------|------------------|------------------|
> > | Imbalanced Data (Reweighting) | 36.12            | 0.0201           | 0.6741               | 0.3677            | 6.40 $\pm$ 0.09 | 20.00            | 0.5208 $\pm$ 0.0025 |
> > | Imbalanced Data               | 35.77            | 0.0205           | 0.6840               | 0.3240            | 6.31 $\pm$ 0.06 | 22.11            | 0.5205 $\pm$ 0.0019 |
> > | + Bias Loss                   | 35.18            | 0.0204           | 0.6868               | 0.3204            | 6.31 $\pm$ 0.09 | 22.16            | 0.5173 $\pm$ 0.0037 |
> >
> > $\newline$
> >
> > 2. Quantify the model coverage loss:
> >
> > We thank the reviewer for their suggestion. We have already incorporated the precision and recall scores when re-training the models on the synthetic datasets in Table 6 and Table 3 for the FFHQ and CIFAR-10 dataset respectively. To further highlight these values we have moved Table 6 from the appendix to the main text.
> >
> > $\newline$
> >
> > Thank you again for your thoughtful review. We made a significant effort to address your questions, including new experiments and paper edits. Please let us know if you have additional questions we can address.

---

### Review · Reviewer_CBs8 · 2024-06-11

**Summary Of Contributions:**

This paper addresses the challenge of generating unbiased samples from trained GAN models. Unlike previous methods that rely on the availability of an unbiased reference dataset, this approach generates training data from the existing GAN model itself. The process involves assuming the existence of a classifier model and performing a breadth-first search in the latent space to create a suitable dataset. This unbiased dataset is then used to retrain or fine-tune the original GAN model by incorporating additional bias-adjusting terms into the primary loss function.

**Audience:**

Yes

**Broader Impact Concerns:**

No ethical concerns

**Claims And Evidence:**

Yes

**Requested Changes:**

Look at the weaknesses.

**Strengths And Weaknesses:**

Strength:
When there is no access to the training dataset of a GAN, the proposed method is an easy way to overcome the bias existing in the generated sample.



Weaknesses:
Assuming that we have a classifier when the number of classes is large is restrictive and to train such a classifier with acceptable accuracy we need a large labeled dataset.
There are a bunch of quantities whose sizes are not discussed well in the paper:
1- How big should be the large of generated balanced data set?
2- How you compute the \lambda_i in the loss function
 3- How big the mini-batch size should be when you approximate the fairness term in the loss? If the mini-batch is not representative especially when D is large, it could make the training unstable.

It seems that your approach needs the latent space to be disentangled like StyleGAN. Since your approach is for general GANs can you explain what would happen if this doesn’t hold?

---

> ### Author Response · Authors · 2024-07-18
> **Response to Reviewer CBs8**
>
> We thank the reviewer for their time and valuable comments. We appreciate that the reviewer has acknowledged the simplicity and the relevance of our proposed approach when access to the training dataset is not available. Below, we address the concerns raised to further clarify our work:
>
> $\newline$
>
> ### Assuming that we have a classifier when the number of classes is large is restrictive and to train such a classifier with acceptable accuracy we need a large labeled dataset.
>
> This is indeed a limitation. However, for many attributes, particularly those that are of public interest, there exist good classifiers. On the other hand, the reliance on a classifier means that our approach is not reliant on an existing balanced reference set. As far as we know, all existing effective bias mitigation methods require either a classifier or a balanced reference set.
>
> $\newline$
>
>
> ### There are a bunch of quantities whose sizes are not discussed well in the paper:
> ### 1- How big should be the large of generated balanced data set?
>
> In our experiments, we have used synthetic datasets with 50,000 images per class. This is generally kept larger than the original dataset so as to incorporate the diversity that was present in the original dataset. We have mentioned further details about this in section 2.5 Datasets and Implementation Details.
>
>  $\newline$
>
> ### 2- How you compute the \lambda_i in the loss function
>
> Thank you for pointing this out. As mentioned in the paper, the \lambda_{d_i} parameter is inversely proportional to the dataset biases. Thus this value is computed similar to the density ratio estimation scaling factor, i.e. $\lambda_{d_i}$ = $\frac{1/|D|}{p_{bias}(d_i)}$. This allows us to give more importance to the biases from under-represented classes. We have added this equation in the paper to clarify this point.
>
> $\newline$
>
> ### 3- How big the mini-batch size should be when you approximate the fairness term in the loss? If the mini-batch is not representative especially when D is large, it could make the training unstable.
>
> Since we compute pseudo fairness values per batch, we require the batch size to be at least equal D. Thus, in this case for a fair generative model we can expect the loss to be approximately equal to 0 and we can expect the training to be stable and effective. However, when D is larger, and computational resources are limited, we will have to aggregate multiple mini-batches when computing the loss function. Thus, we can update the model weights after a specific number of mini-batches. This will handle any instability that could have risen. In our experiments, we have maintained a batch size of 256 for the CIFAR-10 dataset and 32 for the FFHQ dataset.
>
> $\newline$
>
> ### It seems that your approach needs the latent space to be disentangled like StyleGAN. Since your approach is for general GANs can you explain what would happen if this doesn’t hold?
>
> Yes, that is correct, we make an assumption that the latent space of the generative model should be disentangled but not “perfectly” disentangled to allow for fair image generation which serves as the synthetic dataset for further training. It is important to note that most well trained generative models would have a disentangled latent space. Additionally, we have used the Z subspace in the StyleGAN model instead of the W+ latent space, which is generally not expected to be “perfectly” disentangled since it is easier to control image quality using that latent space. Thus, we expect the approach to be widely applicable to most generative models including diffusion models which have semantically disentangled latent spaces.
>
>
> Thank you again for your thoughtful review. We made a significant effort to address your questions, including paper edits. Please let us know if you have additional questions we can address.

---

### Review · Reviewer_kb2s · 2024-07-08

**Summary Of Contributions:**

The work introduces a method to create an unbiased GAN from a biased one by rebalancing the model distribution, generating balanced data with an evolutionary algorithm, and proposing a bias mitigation loss function. It shows improved fairness and maintained image quality on StyleGAN2 and CIFAR10 datasets.

**Audience:**

Yes

**Claims And Evidence:**

Yes

**Requested Changes:**

Fig number is missing on page 3.
Many notations are not defined, e.g., the ratio w(x_i) is not properly introduced and defined. Additionally, the writing needs to be significantly improved; the mathematical parts (such as Sec 2.2 and 2.3.2) lack rigor and include many approximate results.

**Strengths And Weaknesses:**

The main weakness: The novelty in this paper is limited, and the writing needs to be significantly improved (see below).

---

> ### Author Response · Authors · 2024-07-18
> **Response to Reviewer kb2s**
>
> We would like to thank the reviewer for their time and valuable comments. We have addressed the reviewers' concerns below and have updated the paper based on their review as necessary.
>
>
> $\newline$
>
> ### Fig number is missing on page 3.
>
>
> Thank you for pointing that out, we have fixed the missing figure number on page 3.
>
>
> $\newline$
>
> ### Many notations are not defined, e.g., the ratio w(x_i) is not properly introduced and defined.
>
>
> The term $w(x)$ is defined as $\frac{1/|D|}{p_{bias}(C(x))}$ = $\frac{1/|D|}{|C(x)|/|N|}$ where C is the auxiliary classifier; $C(x)$ $\in$ D is the predicted class for the sample x such that $|C(x)|$ is the number of samples present in the dataset for class $C(x)$ and $|N|$ is the total number of samples in the dataset. We have updated the paper and have added this definition.
>
> $\newline$
>
> ### Additionally, the writing needs to be significantly improved; the mathematical parts (such as Sec 2.2 and 2.3.2) lack rigor and include many approximate results.
>
>
> For the baseline method in section 2.2, we have adapted the approach proposed by Choi et al [1] for our use case. Thus the approximations present in the paper directly follow from their work. We would highly appreciate it if the reviewer could point out other locations where the notations or the text was not clear and we can modify and improve it accordingly.
>
> $\newline$
>
> Thank you again for your thoughtful review. We made a significant effort to address your questions, including paper edits. Please let us know if you have additional questions we can address.
>
>
> [1] Choi et al. "Fair Generative Modeling via Weak Supervision" ICML 2020

---

### Decision · Action_Editor_Vjwq · 2024-08-26

**Recommendation:** Reject

**Comment:**

This paper proposes an approach to construct a fair GAN from an existing biased GAN by generating a balanced dataset from the biased GAN and fine-tuning on it. In addition to the use of balanced synthetic dataset, a novel bias mitigation loss is proposed. Experimental results on FFHQ and CIFAR10 with StyleGAN2 show that the proposed method greatly improves the fairness while maintaining the image quality.

Overall, the paper is easy to understand, and the proposed method seems to be technically sound. The authors also address many concerns and issues raised by the reviewers. However, the explanation and analysis of the main algorithm about the generation of the balanced dataset are still lacking, so the main contribution seems to be insufficiently supported. For example, the description on the proposed evolutionary algorithm to generate the balanced data in Algorithm 2 is very lacking, and moreover there is no in-depth analysis on this algorithm as well as the generated data, even though most of the performance improvement is attributed to this. How can you ensure the sufficient diversity and mode coverage for underrepresented classes from this kind of self-augmenation of the biased GAN? Also, the proposed method seems to be highly relying on the classifier, however there is no analysis on the influence of the classifier, e.g., what if we use the classifier that is biased differently from the generative model?

More clarification and analysis on the main algorithm would be necessary, and based on recommendations of the reviewers, I would recommend the paper for rejection. I would recommend the authors to resubmit the paper after doing some modifications on it.

**Audience:**

The use of balanced synthetic dataset for making a fair GAN can receive some interest from TMLR's audiences since it is not necessary to access to the original dataset or the reference dataset, which would be more practical.

**Claims And Evidence:**

This paper proposes a method for making a fair GAN from an existing biased GAN by generating a balanced dataset from itself and then fine-tuning on the generated dataset with a novel bias mitigation loss. Experimental results show that the proposed method significantly improves the fairness while maintaining the image quality.

However, the explanation and analysis of the main algorithm about the generation of the balanced dataset are insufficient. There is no in-depth analysis on this algorithm as well as the generated data, even though most of the performance improvement is attributed to this. Also, there is no analysis on the influence of the classifier that would be a key component in the proposed algorithm.

**Resubmission Of Major Revision:**

The authors may consider submitting a major revision at a later time.